# High-Intensity Laser Therapy (HILT) as an Emerging Treatment for Vulvodynia and Chronic Musculoskeletal Pain Disorders: A Systematic Review of Treatment Efficacy

**DOI:** 10.3390/jcm11133701

**Published:** 2022-06-27

**Authors:** Małgorzata Starzec-Proserpio, Marcela Grigol Bardin, Julie Fradette, Le Mai Tu, Yves Bérubè-Lauzière, Josianne Paré, Marie-Soleil Carroll, Mélanie Morin

**Affiliations:** 1Department of Midwifery, Centre of Postgraduate Medical Education, 01-004 Warsaw, Poland; m.starzec@outlook.com; 2Department of Obstetrics and Gynecology, School of Medical Sciences, Campinas University, São Paulo 13083-887, Brazil; bardinmarcela@gmail.com; 3School of Rehabilitation, Faculty of Medicine and Health Sciences, Université de Sherbrooke, Research Center, Centre Hospitalier Universitaire de Sherbrooke (CHUS), Sherbrooke, QC J1H 5N4, Canada; julie.fradette@usherbrooke.ca; 4Urology Service, Department of Surgery, Faculty of Medicine and Health Sciences, Université de Sherbrooke, Research Center, Centre Hospitalier Universitaire de Sherbrooke (CHUS), Sherbrooke, QC J1H 5N4, Canada; le.mai.tu@usherbrooke.ca; 5Department of Electrical and Computer Engineering, Université de Sherbrooke, Sherbrooke, QC J1K 2R1, Canada; yves.berube-lauziere@usherbrooke.ca; 6Department of Obstetrics and Gynecology, Faculty of Medicine and Health Sciences, Université de Sherbrooke, Research Center, Centre Hospitalier Universitaire de Sherbrooke (CHUS), Sherbrooke, QC J1H 5N4, Canada; josianne.pare@usherbrooke.ca; 7Research Center, Centre Hospitalier Universitaire de Sherbrooke (CHUS), Sherbrooke, QC J1H 5N4, Canada; marie-soleil.carroll@usherbrooke.ca

**Keywords:** pain management, vulvodynia, high-intensity laser therapy, musculoskeletal disorders, chronic pain

## Abstract

High-intensity laser therapy (HILT) has been gaining popularity in the treatment of chronic musculoskeletal pain, including vulvodynia. The objective of this study was to critically appraise and synthesize the available evidence on the efficacy of HILT for reducing pain and improving function in vulvodynia and other chronic primary musculoskeletal pain conditions. Electronic databases and the grey literature were searched. Effects on pain intensity, function, and adverse events were assessed. One study investigating HILT in the treatment of vulvodynia and 13 studies on the treatment of chronic musculoskeletal pain were selected. The study assessing vulvodynia showed favorable results for reducing pain. Regarding chronic musculoskeletal pain, 12 out of the 13 studies selected consistently showed that HILT was more effective than the placebo/active comparator for reducing pain and improving function. The available effect sizes for pain showed large to huge effects. Similar effects were observed for function except for two studies showing moderate effects. The GRADE score was moderate. Conclusions: There are insufficient data to support the use of HILT in vulvodynia, but the promising results encourage further research. HILT appears to be effective in musculoskeletal pain conditions. More high-quality studies are needed to identify effective laser protocols.

## 1. Introduction

Musculoskeletal pain is the leading cause of disability internationally and a major societal burden [1]. The International Association for the Study of Pain (IASP) has placed chronic musculoskeletal pain among the six chronic pain sub-groups [2]. Vulvodynia, a neglected condition with lifetime estimates ranging as high as 10–28% [3], is often proposed to fall under the chronic musculoskeletal pain domain given the involvement of pelvic floor muscles (PFM) [4]. The etiology of vulvodynia points toward multifactorial contributing mechanisms and is not related to an identified disease [5]. Similar to other chronic musculoskeletal pain disorders, common pathways include psychosocial, muscular, inflammatory, and neuroproliferative factors [4,6,7]. Of these, altered pelvic floor muscle contractility and increased tension were shown in women with vulvodynia compared to asymptomatic controls [4]. It has been argued that these muscle dysfunctions in women with vulvodynia could be either a cause (e.g., muscle tension stimulates endomuscular nociceptors, which could lead to referred pain and central sensitization) or a consequence of pain (e.g., a protective-like reaction in response to a painful stimulus or psychological factors, such as the fear of pain) [8,9]. Moreover, an inflammatory process has also been described as playing a role in the pathophysiology of vulvodynia [7] and chronic musculoskeletal pain [10]. As an explanation for pain chronification, it has been suggested that inflammation could trigger neuroproliferation, which could, in turn, lead to hyperalgesia and allodynia [11]. Given these shared pathophysiological mechanisms, similar treatment approaches are often proposed for chronic musculoskeletal pain and vulvodynia [6,12].

Several treatment modalities are available for treating vulvodynia [12]. Although they provide some improvement, effective treatments are still limited and pain persists for a large proportion of women [12]. In the 1960–1970s, laser therapy was proposed for the treatment of various painful conditions [13,14], namely, chronic pain conditions, and it continues to spark the interest of clinicians and researchers. More recently, High-Intensity Laser Therapy (HILT), a Class IV laser (over 500 mW), was introduced in physical therapy and gynecology settings. With the increase in power and the use of different wavelengths, HILT is suggested to have higher penetrative capacities to treat deeper tissues compared to low-level laser therapy (LLLT) [15]. Moreover, the higher dosage of irradiation is suggested to result in enhanced anti-inflammatory, anti-nociceptive, and muscle-relaxing effects, which is consistent with the underlying pathophysiological mechanisms of both chronic musculoskeletal pain and vulvodynia [15,16,17]. These promising advantages of HILT yielded a growing interest in the use of lasers for the treatment of vulvar pain. The Food and Drug Administration (FDA), International Urogynecological Association (IUGA), and International Continence Society (ICS) were therefore compelled to issue a warning statement to investigate the evidence supporting this modality [18,19]. There is currently no systematic review investigating the efficacy of HILT in vulvodynia nor the quality of the evidence available. Moreover, extending our review to include chronic musculoskeletal pain conditions sharing similar pathophysiological pathways is relevant for gathering evidence on the selection of laser parameters and dosimetry, and thereby guides the development of a vulvodynia treatment protocol. To this end, we targeted conditions that fall into the chronic primary pain domain (not directly attributable to a known disease or damage process) as proposed by the IASP Taskforce for the Classification of Chronic Pain [20]. Conditions arising from an underlying disease such as spondylosis, radiculopathy, or osteoarthritis are not included within this group, but fall under other pain categories, e.g., chronic secondary musculoskeletal pain or chronic neuropathic pain [20]. That is why the available reviews for HILT in chronic musculoskeletal pain cannot serve our purpose as they encompassed heterogeneous samples, including patients with radiculopathy, nerve compression, subacromial impingement syndrome, and frozen shoulder [21,22], which all have etiologies distinct from vulvodynia. Moreover, these conditions may also be a significant confounder in the assessment of HILT efficacy. Additionally, a comprehensive investigation of the laser parameters and dosimetry is warranted since these directly influence the biological effects on the tissues and, therefore, the efficacy of treatment for reducing pain and improving function.

### Objectives

The aim of this systematic review was thus to systematically locate, critically appraise, and synthesize the available evidence on the effectiveness of HILT in reducing pain and improving function in (1) women with vulvodynia and (2) patients with chronic primary musculoskeletal pain. Moreover, laser parameters were also retrieved, and effect sizes calculated, whenever possible, to discuss their relative effects.

## 2. Materials and Methods

This systematic review was reported in accordance with Preferred Reporting Items for Systematic Reviews and Meta-Analyses (PRISMA) guidelines [23]. The protocol was registered on the International Prospective Register of Systematic Reviews (PROSPERO; identification number CRD42018112399). The WALT (World Association of Laser Therapy) standard was used to design and carry out the systematic review [24].

### 2.1. Identification and Selection of Trials

The following bibliographic databases were searched from inception to 9 December 2021: Amed, CINAHL, psyInfo and Sportdiscuss via EBSCO, Medline via OVID, Embase via Scopus, and Cochrane CENTRAL via Cochrane library. Grey literature was searched in Proquest, clinicaltrials.gov, IPPS congress, ISSWHS congress, and the WALT congress. A combination of keywords related to the cited conditions (vulvodynia, musculoskeletal disorders, myofascial pain), intervention (HILT), and population (chronic pain patients) was searched (see Appendix A for more details on the search procedures). The literature search results were uploaded into the EndNote software program. Two reviewers (M. S-P. and M. GB) independently screened the titles and abstracts and identified relevant articles that met the inclusion criteria. Full-text articles for all the titles that appeared to meet the eligibility criteria were obtained and then assessed for eligibility. Disagreements were resolved through discussion and reasons for exclusion were recorded.

### 2.2. Eligibility Criteria

Studies were selected according to the criteria outlined in Table 1. Eligibility criteria for chronic primary musculoskeletal pain disorders were selected according to their underlying etiological mechanisms, including non-specific low back pain and myofascial involvement (Table 1). For instance, the trials evaluating laser in acute pain and impingement, or nerve entrapments were excluded as these conditions present a distinct etiology from vulvodynia. As for the laser intervention more specifically, studies evaluating HILT as the primary intervention were included. All of these emit more than 500 mW of laser light and are thus considered Class IV. For the purpose of this review, we opted to investigate high-power lasers with no ablative or tissue necrotizing effects. Therefore, we searched laser systems with emission modalities that allow for the control of photothermal and photomechanical processes to obtain therapeutic effects without tissue damage. Based on that reasoning, CO_2_ and Erbium:YAG lasers were excluded.

### 2.3. Data Extraction and Management

Two independent reviewers (M.S.P. and M.G.B.) extracted the relevant information from the full text using a standardized Excel form that was piloted prior to use. The following information was extracted: study sample, subject demographics, HILT details, data collection time points, outcome measures, adverse events, dropouts, and results. To mitigate for the missing data, physics formulas were used to calculate the unreported laser parameters. The following formulas were used based on a previous review on laser therapy [25].
Energy dose (J) = average power in Watts (W) × exposure time in seconds (s);
Energy density (J/cm^2^) = average power in Watts (W) × exposure time in seconds (s)/area of treated surface or probe tip (cm^2^);
Average power density (W/cm^2^) = average power in Watts (W)/area of treated surface or probe tip (cm^2^)

To calculate the average power for pulsed lasers, an additional formula was used: Average power = peak power (W) × pulse duration (s) × frequency (Hz) [26].

When information on laser parameters was missing, but the type and model of the laser were mentioned, the manufacturer’s website was used to obtain the required data. In studies that used scanning and stationary (point-to-point) applications, the parameters for each application mode were calculated and reported separately based on the available data.

### 2.4. Data Analysis and Quality of Evidence

Data were aggregated and a narrative critical analysis was performed based on the included studies. Descriptive data were used to characterize the study population. Summary data for each type of HILT were provided and a narrative report on the findings was completed. In addition, the magnitude of the differences between the treatment groups was calculated using Cohen’s *d* effect size whenever possible. These coefficients, characterizing between-group differences, were considered only for studies with a normal distribution of data. Effect sizes were interpreted as weak (0.2–0.49), medium (0.5–0.79), large (0.8–1.19), very large (1.2–1.99), and huge (≥2) [27,28] to assess the additional or differential effect of HILT. A narrative synthesis of the quality of evidence was performed using the GRADE system [29,30,31].

### 2.5. Assessment of Risk of Bias

Two authors (M. S-P. and M. GB) independently screened all the selected studies assessing the risk of bias. Disagreements were discussed until a consensus was reached. When needed, a third researcher (M.M.) became involved. Risks of bias for non-randomized studies were evaluated with the ROBINS-I tool. The first two domains cover confounding and selection of participants into the study. The third domain addresses classification of the interventions themselves. The other four domains address issues after the start of interventions: biases due to deviations from intended interventions, missing data, measurement of outcomes, and selection of the reported result [26]. Risks of bias for randomized clinical trials (RCTs) were assessed using the Revised Cochrane Collaboration tool (risk of bias RoB 2.0), which covers the following: randomization process (allocation sequence generation and concealment), blinding, incomplete outcome data, measurement of the outcome (blinding of outcome assessors), and selective outcome reporting [32]. Potential bias was evaluated as having a low risk of bias, some concerns, or a high risk of bias for each of the 5 domains. An overall decision on the risk of bias was then made for each study according to the following criteria: (1) a study was considered as having an overall high risk of bias when high risk was attributed to at least one domain or when some concerns of bias were attributed to multiple domains; (2) a study was thought to have some concerns of bias when any of the 5 domains was rated as having some concerns of bias; (3) a study was deemed to be at low risk of bias when all domains were scored as low risk.

## 3. Results

### 3.1. Search Results

The search yielded 410 studies pertaining to vulvodynia once duplicate studies were removed (see Figure 1 for the flow diagram). Of these, 405 studies were excluded because they failed to meet the eligibility criteria. Five studies were read in their entirety and were excluded (three studies did not use HILT and one study did not specify the type of laser), resulting in only one relevant study being included in this systematic review [33]. Figure 1 shows the flow of vulvodynia studies.

Regarding musculoskeletal pain studies, 2559 trials were retrieved from the search after the removal of duplicates (Figure 2). Of these, 2528 studies were excluded because they failed to meet the eligibility criteria. Thirty-one studies were read in full and eighteen were excluded (thirteen were not HILT, and five did not meet the eligibility criteria for the study population). This resulted in 13 relevant studies included in this systematic review [34,35,36,37,38,39,40,41,42]. Table 2 presents a summary of the characteristics of the included studies and Figure 2 shows the flow of musculoskeletal studies within the search process.

### 3.2. Vulvodynia Study Summary

Only one retrospective vulvodynia study was included in this systematic review [33] and it was considered as having a high risk of bias in six out of seven domains assessed (Figure 3). Thirty-seven women with a mean age of 32.9 were treated with 532 nm of HILT, with an energy density of 10 J/cm^2^, over an average of 2.84 sessions (range: 1–8). The mean time between HILT sessions and data collection was 2.8 years (range: 1–6). Participants were divided into the following groups: HILT only (n = 22), HILT before surgery (n = 13), and surgery followed by HILT (n = 2). Patients were evaluated using a 17-item survey designed by the authors. The survey questions were designed using a 5-point Likert scale and evaluated vestibular pain, sexual pain, sexual quality of life, and satisfaction with treatment. Although pain was measured before treatment, no visual analogue scale (VAS) measurement was taken after the intervention. Participants were rather asked to report their subjective reduction in pain using a three-level categorial scale (i.e., more pain, no change, or less pain). Overall, 68% of the participants reported a decrease in the pain associated with vestibulodynia after undergoing HILT. Most of the participants (63%) expressed satisfaction with HILT as evaluated with a dichotomous question (i.e., satisfied/not satisfied), and 38% stated that they would recommend laser therapy to a friend with the same problem. It is also worth noting that no between-group statistics were computed and that these data were derived from the merged assessment of all groups. Only one woman reported an increase in symptoms after undergoing laser therapy. No other adverse events were reported.

### 3.3. Characteristics of Included Musculoskeletal Pain Studies

Of the musculoskeletal pain studies included, we found nine studies that investigated chronic low back pain [34,37,38,39,41,42,43,44,45], two trials that focused on chronic neck pain [35,36], one study that examined myofascial pain syndrome of a trapezius muscle [40], and one study that assessed myogenic temporomandibular joint disorder [46]. Table 2 provides a summary of patient characteristics, sample sizes, co-interventions and comparators, outcome measures, and duration of follow-up. The studies included 726 participants in total at baseline (range 20–76) with ages ranging from 18 to 65 years old. The mean or median pain duration at baseline was reported in 9 of the 13 studies and ranged from 5 months to 2.5 years [34,35,36,37,40,43,44,45,46]. Three studies recorded dropouts. In a study by Abdelbasset et al. [44], one person from the sham HILT group discontinued. In a study by Ekici et al. [46], two patients in the HILT group and one patient in the sham HILT group dropped out before completing the study. In the study by Basford et al. [37], 2 participants missed one of the 12 sessions, and 3 participants did not show up for the 1-month follow-up evaluation. The information about their group allocation was missing. The reasons for dropping out were not mentioned in these studies.

### 3.4. Quality Assessment/Risk of Bias in the Included Musculoskeletal Pain Studies

Figure 4 depicts the risk of bias determined for each domain. As regards the overall risk of bias for each study, three studies were considered as having a low risk of bias, four studies as having some concerns, and six other studies were assessed as having a high risk of bias. Only four studies described the concealment process in detail, such as the use of numbered envelopes [40,44,46] or folded cards [41]. All remaining studies [34,35,36,37,38,39,42,43,45] failed to report on allocation concealment. Therefore, all nine of those studies were assessed as having some concerns of bias arising from the randomization process. Six studies were deemed as having a low risk of bias due to deviations from the intended intervention [35,36,37,40,44,46]. The remaining seven studies were declared as having some concerns of bias because the participants were or could be aware of the intervention groups and because there was no information on whether there were deviations from the treatment protocol that were likely to impact outcomes. All but one trial [39] were assessed as having a low risk of bias due to missing outcome data. The study by Conte et al. [39] was classified as having some concerns because the authors presented the results of 25 patients in each group, while the text states that there were 28 patients per group. Six studies [35,36,37,40,44,46] were assessed as having a low risk of bias in terms of measurement of the outcome data, three as having some concerns of bias [34,41,42], and four as having a high risk of bias [38,39,43,45]. Although the participants in the study by Alayat et al. [34] were blinded, one of the three study groups only used HILT. Therefore, patients could guess which group they were assigned to. The studies conducted by Gocevska et al. [42] and Fiore et al. [41] compared the effects of two different treatment methods; therefore, it was not possible to blind the participants. The outcome measures were self-reported by participants. Therefore, according to RoB 2.0 [32], the participants were considered to be outcome assessors, even if the interviewer administering and filling out the questionnaire was blinded. In the four studies with a high risk of bias [38,39,43,45], the treatment groups received an additional intervention compared to the control group. Thus, the participants could not only guess their group allocation, but could also potentially perceive their treatment to be superior/inferior to the other study group. All of the trials included were deemed to be at low risk of bias in the selection of the reported results since the outcome data were not likely to have been selected based on the results.

### 3.5. Comparators and Co-Interventions

In seven studies, sham laser (SL) was used in the control group [34,35,36,37,40,44,46]. In the six remaining studies, other treatment modalities were used as comparators: ultrasound therapy [41,42], low-level laser therapy [43], pulsed electromagnetic field [45], conservative physical therapy [38], home exercise program [43,45], and back school exercises [39]. Three studies [37,41,46] used HILT alone as an intervention. In the 10 other studies, co-interventions were used and included exercises [34,35,36,39,40,42,43,44,45] or conservative physical therapy modalities [38]. In the study by Alayat et al. [34], both HILT alone and in conjunction with exercises were investigated. Different forms of exercise were used in the selected studies. For presentation clarity, all exercises were grouped together under the term “EXERCISE (EX)”.

### 3.6. Outcomes

#### 3.6.1. Primary Outcome—Pain

A greater decrease in pain intensity in the HILT groups relative to the comparators was observed in all 13 trials assessed with either a numerical rating scale (NRS) or visual analogue scale (VAS) (see details in Table 2). In all the studies, except one trial [43], the superior effect of HILT was observed regardless of the co-intervention or comparison group. In a study by Abdelbasset et al. [43] investigating three arms, the HILT + EX group showed a superior effect compared to the EX group; however, comparison between the LLLT + EX and the HILT + EX groups post-intervention showed no significant differences (*p* > 0.05) as assessed with post hoc analysis. In another study investigating three arms by Alayat et al. [34], it is worth noting that the HILT + EX group showed a superior effect compared to the SL + EX group, while the smallest effect was found in the intervention group involving HILT alone.

#### 3.6.2. Secondary Outcome—Function

In 12 of the 13 trials, significantly more improvement in function was noted in the HILT groups compared to the sham laser/active comparator groups, as assessed with the Roland Disability Questionnaire [34], Modified Oswestry Disability Questionnaire [34,39,45], Oswestry Disability Index [37,38,41,42,43,44], Neck Disability Index [35,36,40], Pain Disability Index [45], and Jaw Functional Limitation Scale-20 [46] (Table 2). It should be noted that in a study by Abdelbasset et al. [43] investigating three arms, the HILT + EX group showed a superior effect compared to the EX group; however, a comparison between the LLLT + EX and the HILT + EX groups post-intervention showed no significant differences (*p* > 0.05) through post hoc analysis.

#### 3.6.3. Time-Dependent Improvements within HILT Groups

Eight studies [35,36,38,39,41,43,44,45] collected data only at pre- and post-treatment time points and included no further follow-up. In the remaining studies that provided follow-ups to participants, the benefits of HILT were shown to persist over time for up to 3 months [34,37,40,42,46]. However, all the included articles presented limited follow-up duration. The longest reported follow-up period assessing the beneficial effects of HILT was 3 months [34,42]. Details are presented in Table 2.

#### 3.6.4. Treatment Efficiency Considering Laser Parameters and Dosimetry

As shown in Table 3, various laser parameters and a large range of doses were used in the included studies. For instance, the studies differed in the mode of HILT application, with some using both a scanning and stationary application [34,35,36,39,40,41,45,46], some using scanning only [42,43,44], and one using stationary only [37]. In the case of one trial, the mode of application was not mentioned [38]. Regarding the use of continuous or pulsed mode, in the majority of the selected studies, pulsed mode HILT prevailed. Only three studies used the continuous wave laser mode [36,37,42]. In the study by Alayat et al. [36], a combination of dual waveforms (continuous and pulsed) was used synchronously. All remaining HILT settings varied greatly among the studies, using different wavelengths (532–1064 nm), energy densities (0.72–266.7 mW/cm^2^ in scanning mode and 2.15–2775 mW/cm^2^ in stationary mode), energy doses (801.6–48,000 J), number of sessions (10–24), duration of sessions (30 s–15 min), etc. Therefore, given the large heterogeneity used in HILT protocols, data could not be pooled, and subgroup analyses could not be performed to determine the most optimal HILT parameters.

#### 3.6.5. Adverse Events

Eight studies [34,35,36,38,39,43,44,45] omitted to report the occurrence or absence of side effects. In the remaining five studies [37,40,41,42,46], there was a clear statement about no adverse events. One trial [37] mentioned a tendency for the HILT group to report a mild “warmth” more often during treatment compared to the sham laser, but this trend did not reach statistical significance.

#### 3.6.6. Effect Sizes and Clinical Significance

The effect size values were available for 11 studies as depicted in Table 4. In one study [36], Cohen’s d effect size values were retrieved from the source article. In 10 studies [34,35,38,39,40,42,43,44,45,46], it was possible to compute the coefficients using the available data. For the study by Ekici et al. [46], we were able to compute the effect size calculations only for pain due to a lack of reporting the total score of the functional outcome (Jaw Functional Limitation Scale-20). Therefore, the effect size values were interpreted from a total of 10 studies [34,35,36,38,39,40,43,44,45,46] for pain and 9 studies [34,35,36,38,39,40,43,44,45] for function as it was possible to investigate either the additional or the differential effect of HILT (i.e., the only factor distinguishing the groups was HILT). To allow the assessment of the differential contribution of HILT, we did not include the studies or groups with different comparators. Therefore, the study by Gocevska et al. [42] was not taken into consideration. For the study by Alayat et al. [34], we looked at the effect size values between the HILT + EX and SL + EX groups only. Similarly, for Abdelbasset et al. studies [43,45], we considered the effect size values between the HILT + EX and EX groups only. For the pain outcome, effect sizes were large to huge in all 10 studies [34,35,36,38,39,40,43,44,45,46] at all time points. Similar effect sizes were observed for function with two exceptions. In the study by Alayat et al. [34], a moderate effect was found for function measured post-treatment by the Modified Oswestry Disability Questionnaire between the HILT + EX and SL + EX groups. In the study by Abdelbasset et al. [44], while a huge effect was observed for function measured post-treatment by the Oswestry Disability Index, a moderate effect was found for measurements by the Pain Disability Index.

#### 3.6.7. Quality of Evidence

The GRADE approach provided the framework to assess the quality of evidence. The GRADE summary of results table is available in Appendix B. Overall, the quality of evidence for pain and functional outcomes was moderate, meaning that the true effect is likely to be close to the estimated effect, but there is a possibility that it could be different [31]. The level of evidence was downgraded due to imprecision that is inherent to a narrative synthesis. Although we have calculated the effect sizes for individual studies, they were not available for all the RCT’s, which prevented us from pooling precise estimates.

## 4. Discussion

The purpose of this systematic review was to critically appraise the literature on the effectiveness of HILT in women with vulvodynia and chronic primary musculoskeletal pain disorders. Only one study was found in women with vulvodynia and, although favorable results were reported regarding HILT effectiveness, the high risk of bias related to this study prevents us from making any recommendations for HILT in women with vulvar pain. Regarding the included musculoskeletal pain studies, HILT was consistently shown to be effective for decreasing pain and improving function compared to the sham laser/active comparators in 12 out of the 13 included studies. These changes were found to be clinically relevant according to the available effect sizes. However, these results should be interpreted with caution, considering that the quality of evidence was scored as moderate (i.e., the true effect is probably close to the estimated effect).

The study by Leclair et al. [33] was the only study identified in our search investigating the efficacy of HILT in women with vulvodynia. Although the results were favorable with a high proportion of women reporting a reduction in pain and being satisfied with the treatment, these findings should be interpreted with caution given the serious risk of bias attributed to the majority of the ROBINS-1 tool domains. Methodological limitations include bias inherent to the retrospective design of the study (i.e., no randomization, group assignment based on each patient’s choice, no blinding, and various follow-up durations) as well as the use of non-validated outcomes for pain, the absence of a sexual function assessment, and concomitant vestibulectomy performed in some participants. Moreover, very limited information on laser parameters was provided, which prevents the replication of the study and the assessment of parameter relevance.

As for musculoskeletal pain studies, despite variations in conditions and treatment comparators, findings concur to support the efficacy of HILT in reducing pain and improving function. Twelve out of the thirteen musculoskeletal pain studies showed a superior effect in the HILT group compared to the comparison groups. Moreover, these favorable results remained significant at the 4-week [37], 9-week [40,46], and 12-week [34,42] follow-up time points. In addition to statistically significant differences, the average changes that occurred after laser treatment both in pain and function surpassed the minimal clinically important differences (MCID) in 12 of the 13 studies. For instance, all the studies, except one [37], showed an average reduction in pain in the HILT group at all time points that exceeded the ≥30% benchmark, reflecting an MCID [47]. Similarly for function, the average changes observed in the laser group were beyond the MCID (i.e., Oswestry Disability Index > 10, [48] Roland Disability Questionnaire > 3.5 [48], Neck Disability Index > 5 [49], and Pain Disability Index > 8.5 to 9.5 [50]) for all studies at all timepoints, except in one study [37]. In the function analysis, we omitted the study by Ekici et al. [46] as there is currently no established MCID value for Jaw Functional Limitation Scale-20 [51]. Notwithstanding the favorable results observed in most of the included musculoskeletal studies, it should be highlighted that further high-quality studies are needed prior to recommending the use of HILT in clinical settings, given that only three trials had a low risk of bias [40,44,46], four studies had some concerns [35,36,37,41], and six trials had a high risk of bias [34,38,39,42,43,45]. When examining subscales for assessing risk of bias, it should be noted that five studies had an increase in their overall risk of bias because of the omission to report the concealment method [35,36,37,43,45]. Another factor that contributed to higher risk of bias scores with the RoB 2.0 tool was the use of patient-reported outcomes. This resulted in a high risk of bias related to the outcome measure domain because a high-risk score is automatically attributed when patients become assessors of their own condition via self-reported outcomes (e.g., to assess pain and function) and when patients cannot be blinded to their allocated interventions. This occurred in the studies by Choi et al. [38] and Conte et al. [39] where HILT was provided in addition to exercises, as well as in the studies by Fiore et al. [41] and Gocevska et al. [42] where HILT was compared to ultrasound. A similar situation took place in the studies by Abdelbasset et al. with three study arms [43,45] where HILT was compared to low-level laser therapy [43], magnetic field [45], and exercises [43,45]. It is worth noting that when comparing HILT with another frequently used treatment modality, it would be unlikely to obtain a lower risk of bias. In addition, using patient-reported outcomes for assessing pain and function is inevitable. This unfavorable risk of bias scoring with the RoB 2.0 tool in non-pharmacological trials using self-reported outcomes should be further discussed. Methodological recommendations should be developed to guide researchers in minimizing bias related to such study designs, and the risk of bias assessment tools should be adjusted accordingly.

### 4.1. HILT and Biological Effects on Tissues and Pain

Despite the favorable results found in the included studies regarding the efficacy of HILT, the exact mechanisms of action of HILT remain not clearly understood. The limited number of animal and human studies investigating various laser parameters (e.g., low/high intensity, pulsed/continuous, etc.) in different conditions hinders our comprehension of the effects of lasers for reducing pain. Among the mechanisms of action proposed, it has been suggested that laser treatment may have anti-inflammatory effects through photobiomodulation mechanisms by altering inflammatory markers (e.g., tumor necrosis factor, interleukin 1B, bradykinin, and prostaglandin) in both animal [52,53,54] and human studies [55]. This mechanism appears relevant to our review as inflammation has been reported in both vulvodynia [7] and other chronic musculoskeletal pain disorders [10]. Moreover, HILT is reported to have a photothermal effect. It was recently shown in the prospective study by Alayat et al. [16] that HILT produced an increase in tissue temperature. The authors suggested that this warming effect may potentially lead to improved muscle relaxation and extensibility of the connective tissue and thus, reduce pain. Both muscles and connective tissues are some of the main targets in the treatment of vulvodynia and chronic musculoskeletal pain. Another mechanism of action described is the analgesic effect of lasers through neural inhibition. Indeed, the systematic review by Chow et al. [56] investigating the effects of laser on mammalian nerves (i.e., animal and human) showed that laser, especially at a higher therapeutic dose, results in an anti-nociceptive effect. It does so by disrupting the cytoskeleton, suppressing conduction velocity, and reducing the amplitude of the action potentials in small-diameter nerve fibers that convey nociceptive stimuli. In turn, this may decrease hyperalgesia and allodynia, which are common to both vulvodynia [57] and chronic musculoskeletal pain [58]. HILT is also suspected to have an analgesic effect through endorphin mechanisms. The study by Laasko et al. [59] observed a dose-dependent effect of laser on the circulating level of beta-endorphin, which suggests that laser may also reduce pain through the central pathway. Therefore, the combined peripheral and central mechanisms of laser may potentially have an influence on pain centralization, which is common to both vulvodynia [60] and musculoskeletal chronic pain conditions [61]. Although these proposed mechanisms of action are relevant to the pathophysiological pathways of vulvodynia and chronic musculoskeletal pain conditions, their relation with the findings of our review remains hypothetical. Further human studies are needed to confirm these potential mechanisms in relation to the various HILT parameters and, most importantly, their relevance in pain mediation.

Furthermore, HILT is hypothesized to have similar properties to LLLT, but with augmented effects due to its higher power. A commonly mentioned advantage of HILT is that, with increased power, the depth of penetration may also increase (ref). Moreover, a higher dosage applied to the tissue would potentially result in an enhanced photomechanical effect [62,63]. Overall, HILT has been suggested to overcome the limits of LLLT, such as the limited penetration inside the tissues and the inability to obtain an efficient photomechanical effect [63]. However, our review revealed conflicting results when comparing HILT and LLLT: Alayat et al. [36] proved HILT to be superior to LLLT, while in the study by Abdelbasset et al. [43], a comparison between the LLLT and HILT groups post-intervention showed no significant differences through post hoc analysis (*p* > 0.05). A well-designed RCT should be conducted to compare the effects of LLLT versus HILT for musculoskeletal pain disorders in order to confirm or infirm the superiority of HILT over LLLT.

### 4.2. Clinical Importance of the Results

In addition to statistical significance, clinical significance was also assessed whenever possible by analyzing the computed effect sizes in 10 studies for pain and 9 studies for function, of the 13 studies included. The treatment effect in reducing pain intensity was found to be huge in three studies [36,38,45], very large in four studies [35,43,44,46], and large in three studies [34,39,40]. Similarly, significant effects were observed for function, with all nine studies showing at least a large effect, except in two trials. In the Alayat et al. study [34], a moderate effect was demonstrated immediately post-treatment (in function, measured by Modified Oswestry Disability Questionnaire), which then improved over time, resulting in a large effect at the 12-week follow-up. In a study by Abdelbasset et al. [44], a moderate effect for function was found when measured by the Pain Disability Index, while a huge effect was observed when measured by the Oswestry Disability Index. Overall, the high effect size values, the exceeded minimal clinically important differences for nearly all studies for both pain and function, and the moderate quality of results (GRADE) suggest that HILT may improve pain and function in chronic musculoskeletal pain conditions.

### 4.3. Methodological Considerations

A common issue encountered in the majority of the included studies was the poor reporting of HILT parameters. None of the included studies fully adhered to the key items recommended by the WALT consensus agreement on the design and conduct of clinical studies using laser therapy and light therapy for musculoskeletal pain disorders [24]. Although HILT was found to be an effective treatment approach showing a significant reduction in pain and improvement in function, it is crucial that studies provide sufficient information on laser parameters to allow replication of findings and knowledge transfer to clinical practice. Extensive variation and heterogeneity in parameters of the selected studies (e.g., pulsed/continuous emission, scanning/stationary delivery, various wavelengths, and a wide range in energy dose) prevent the identification of any trends indicating the most optimal and effective laser parameters. The efficacy of laser therapy depends on the protocol used as the mechanisms of action are intrinsically linked to the laser parameters selected. For instance, the laser penetration depth is associated with laser wavelength [64]. The selection of the optimal wavelength thus appears to be important in targeting the tissue of interest to ensure an adequate accumulated amount of energy in the structures to elicit a photobiological reaction [65]. Furthermore, it has been suggested that pulsed mode could yield better outcomes than continuous laser mode because the photothermal effects can be controlled and limited for patient safety by modulating pulse intensity and frequency [15]. Indeed, Ilic et al. [66] found that pulsed laser light produced no neurological or tissue damage as opposed to a continuous wave (for equivalent power density delivered for the same duration), which caused neurological deficits through neuronal tissue necrosis. Pulsed laser light was also hypothesized to produce photomechanical reactions by loading and unloading the cells, creating mechanical stress that could affect cellular behavior and result in anti-inflammatory and analgesic effects [15]. These potential mechanisms remain theoretical and need to be further investigated in both vulvodynia and musculoskeletal pain conditions, which require, as a prerequisite, proper reporting of the laser parameters employed.

### 4.4. Research Implications and Recommendations for Future Studies

We strongly recommend adhering to the WALT consensus agreement on the design and conduct of clinical studies using laser therapy for vulvodynia and musculoskeletal pain disorders. All of the important HILT parameters and their applications should be clearly outlined, including the laser model and type, wavelength, probe tip, output power (for continuous wave mode) or peak and average power (for pulsed mode), pulsing and pulse duration, pulse frequency, dosage, power density, treatment technique (distance), and treatment time and frequency [67]. Ideally, these parameters would be included for all laser application modes when more than one is used (e.g., stationary and scanning). This will provide clarity in the treatment protocol and allow for a thorough analysis of the HILT parameters used. In the current review, the lack of information and the large heterogeneity of the protocols prevented the pooling of data and investigation into the efficacy according to laser parameters. Increasing, the number of high-quality trials that provide all relevant details on laser parameters would eventually facilitate the determination of the most optimal laser parameters. Moreover, multidisciplinary teams including laser experts are warranted, given that treatment providers are not necessarily familiar with the detailed technical features of lasers and the photobiological properties of laser parameters. The inclusion of laser experts in the research process would help provide a better understanding of the treatment being administered and guide the choice of laser settings that need to be used. We also recommend using the most rigorous study design while adhering to recognized guidelines, including the CONSORT statement [68]. Outcome measures should also be selected according to the IMMPACT recommendations and a vulvodynia task force to capture a range of domains impacted by chronic pain [69,70].

### 4.5. Study Limitations

One limitation of this study is the heterogeneity and poor reporting of the laser parameters employed. Consequently, meta-analyses could not be performed and we had to rely on some assumptions and perform numerous computations to mitigate for missing information. This process may have increased the risk of misinterpretations even though only previously used and well-known formulas were employed [25,26]. Extensive variation in laser treatment protocols precludes sub-group analyses or any identification of trends for determining the most optimal laser parameters. Although the efficacy of laser was shown to persist over time, the longest effects investigated were limited to 3 months post-treatment. Moreover, the limited methodological quality of the included studies prevented us from drawing firm conclusions on the effects of HILT in vulvodynia and chronic musculoskeletal pain. Allocation concealment and blinding were frequent issues in the available studies, thus increasing the risk of bias. Admittedly, blinding participants and personnel is difficult in non-pharmacological trials. However, attempts to reduce performance bias using sham laser and treatment comparators that allow for blinding should be considered.

## 5. Conclusions

Findings regarding the efficacy of HILT in vulvodynia, retrieved from only one small retrospective study with a high risk of bias, are insufficient for recommending its use in clinical settings. However, these results encourage conducting further research on HILT in this population. Regarding HILT for other chronic primary musculoskeletal pain conditions, the findings derived from available RCTs consistently showed that HILT was effective in reducing pain and improving function with a large to huge effect size. Given the overall moderate quality of evidence, more high-quality studies are warranted, and these should comprehensively report laser parameters in order to investigate optimal laser protocols.

## Figures and Tables

**Figure 1 jcm-11-03701-f001:**
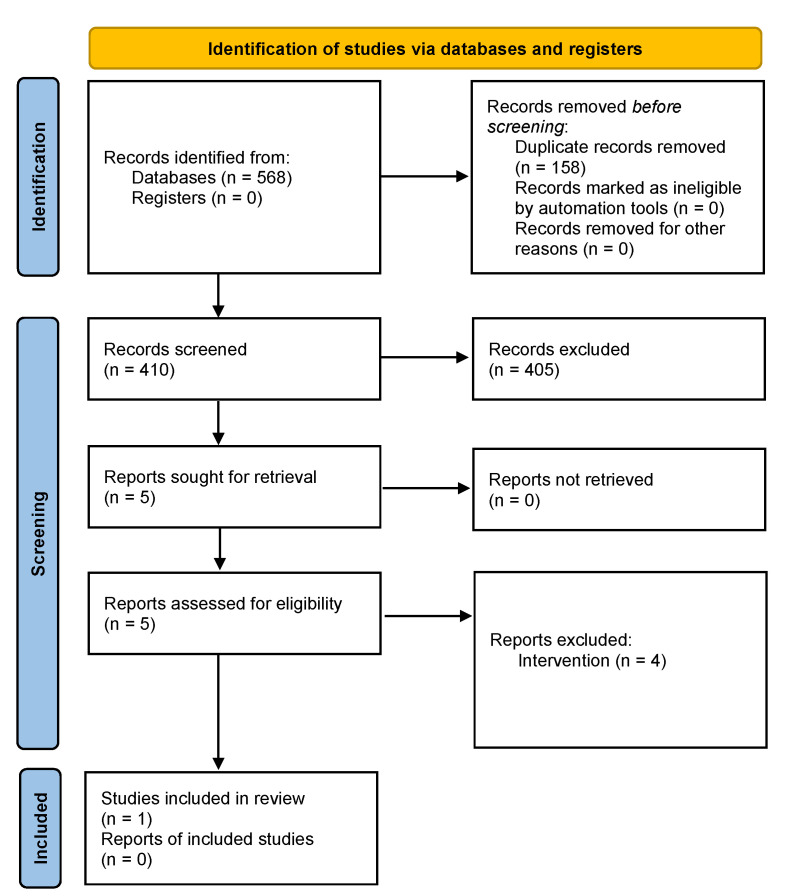
Flow diagrams of literature review for vulvodynia. This figure was adapted from the Preferred Reporting Items for Systematic Reviews and Meta-Analyses (PRISMA) statement.

**Figure 2 jcm-11-03701-f002:**
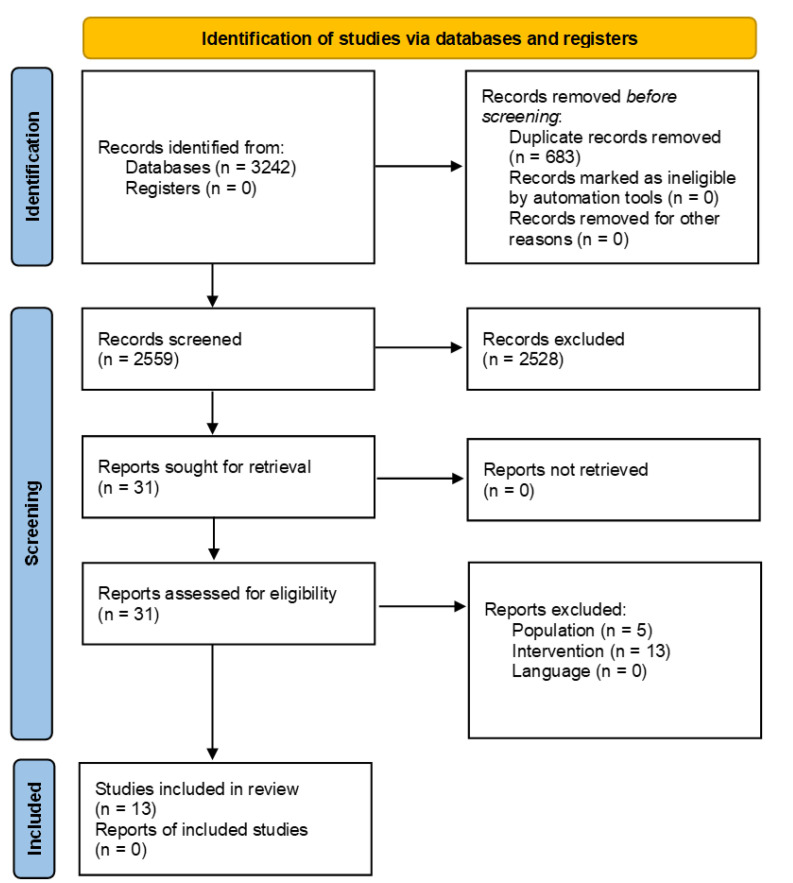
Flow diagrams of literature review for musculoskeletal pain studies. This figure was adapted from the Preferred Reporting Items for Systematic Reviews and Meta-Analyses (PRISMA) statement.

**Figure 3 jcm-11-03701-f003:**
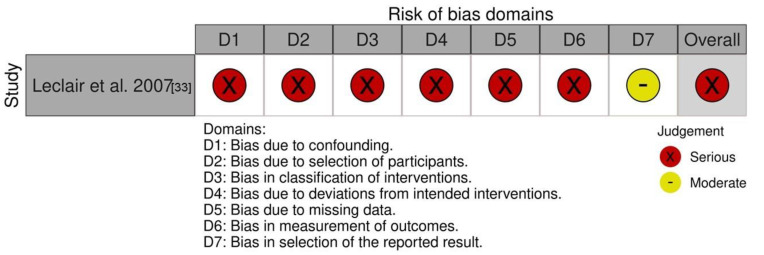
Risk of bias (RoB) of the included vulvodynia study assessed with the ROBINS-I.

**Figure 4 jcm-11-03701-f004:**
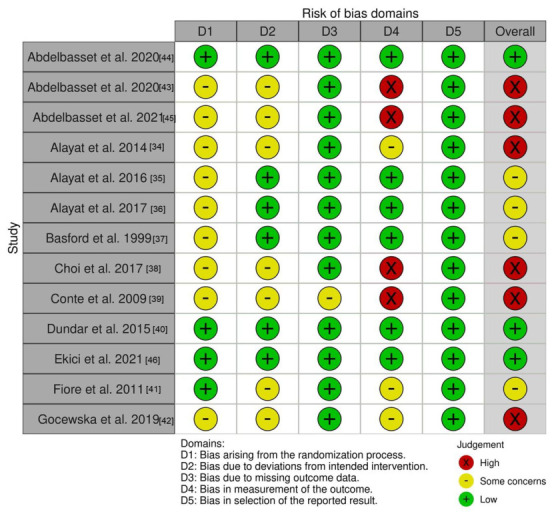
Risk of bias (RoB) of included musculoskeletal pain studies assessed with ROB2.0.

**Table 1 jcm-11-03701-t001:** Eligibility criteria.

Variables	Criteria
Population	Inclusion: -Vulvodynia population: studies involving women with vulvodynia or superficial dyspareunia. -Chronic primary musculoskeletal pain population: Eligible studies comprised (1) nonspecific chronic low back pain, (2) neck pain, or (3) myofascial pain and myofascial trigger point pain.
	Exclusion: -Vulvodynia population: studies involving minor women, pregnant women, women who have undergone organ or bone marrow transplants, or women with other pelvic pain conditions, such as chronic pelvic pain different than vulvodynia, endometriosis, sexually transmitted infections, other vulvovaginal infections, cancer, dermatologic conditions, genitourinary syndrome of menopause (including vulvo-vaginal atrophy), or deep dyspareunia. -Chronic primary musculoskeletal pain population: studies involving participants with widespread musculoskeletal pain (e.g., fibromyalgia), systemic illness/inflammatory condition (e.g., rheumatoid arthritis), or headache. Excluded trials were those also examining patients with low back pain due to specific pathological entities including: (1) specific spinal pathology, (i.e., spondylosis, infection, tumor, osteoporosis, fracture including spondylolisthesis), structural deformity (including scoliotic deformities), inflammatory disorder, or (2) neurological encroachment (radicular or cauda-equina syndrome). *
Intervention	Studies evaluating HILT therapy as the primary intervention were included.
Comparator	Studies including co-interventions were allowed if applied equally to both laser and control groups.
Outcomes	(1) Pain intensity (e.g., pain during intercourse), (2) Functional disability (e.g., sexual function), (3) Participant’s perceived improvement. We also looked at adverse events (worsening of pain, dropouts).
Timing	There were no restrictions based on the length of follow-up of outcomes.
Setting	There were no restrictions based on type of setting.
Design	Given the limited literature available on the effectiveness of laser treatment in women with vulvodynia, randomized controlled trials (RCTs), prospective and retrospective cohorts, case reports, and study protocols were included in the review. Only RCTs were included for the musculoskeletal population.
Language	There were no language restrictions.

* Studies with participants presenting with a neuropathic/radicular component were included if they represented a small proportion of the sample.

**Table 2 jcm-11-03701-t002:** Characteristics of the studies included.

Studies	Sample Size: N Total, Gender (M/F)	Mean Age (Years)	Study Groups	Outcomes	Time Points	Relevant Results
Vestibulodynia						
Leclair et al., 2007 [34]	37 0/37	33	(1) HILT (2) HILT then surgery (3) Surgery then HILT **Comparator**: N/A	17-item survey designed by the authors	2.8 yFU (1–6)	68% of participants reported a decrease in the pain associated with vestibulodynia after HILT treatment. 63% expressed satisfaction with HILT treatment, and 38% stated that they would recommend laser therapy to a friend with the same problem.
**Nonspecific Chronic Low Back Pain**						
Abdelbasset et al., 2020 [45]	35 22/13	40 HILT 39 SL	(1) HILT + EX (2) SL + EX **Comparator**: SL	VAS, ODI PDI	Pre Post	Significant improvements in pain and function were observed in the HILT + EX group, whereas the SL group showed no significant changes. Comparison between the groups post-intervention: The HILT + EX group scored significantly better in comparison to the SL + EX group for pain and function.
Abdelbasset et al., 2020 [44]	60 42/18	32 LLLT 34 HILT 33 EX	(1) HILT + EX (2) LLLT + EX (3) EX **Comparators:** (1) LLLT Infrared, 850 nm laser with 800 mW power, 30 min/session, delivering 1200 J (2) EX home exercise training: strengthening exercises for back and abdominal muscles, stretching exercises for back muscles, at least twice per week	VAS ODI	Pre Post	Significant improvements were observed in the HILT + EX and LLLT + EX groups in both pain and function, whereas the EX group showed no significant changes. Comparison between the three groups post-intervention: a significant difference in all outcome measures. Comparison between the LLLT + EX and HILT + EX groups post-intervention: no significant differences through post hoc analysis.
Abdelbasset et al., 2021 [46]	51 Nr	36 HILT 36 EMF 37 EX	(1) HILT + EX (2) EMF + EX (3) EX **Comparators**: (1) EMF: 30 Hz pulse frequency, for 30 min/session, delivering 14 µT (2) EX: home exercise program: abdominal, back, pelvic muscle stretching, flexibility, mobility and strengthening, 3 x/week	VAS, MODQ, PDI	Pre Post	Significant improvements were observed in the HILT + EX and EMF + EX groups in both pain and function, whereas the EX group showed no significant changes. Comparison between the groups post-intervention: The HILT + EX group scored significantly better in comparison to the EMF + EX group for pain and function. Within-group percent of change for pain and function were greater in the HILT + EX group when compared to the EMF + EX group.
Alayat et al., 2014 [35]	72 72/0	33	(1) HILT + EX (2) SL + EX (3) HILT **Comparator**: SL	VAS, RDQ, MODQ	Pre, Post, 12 wFU	Significant improvements in both pain and function were observed in all 3 groups post-treatment and the results remained consistent at 12 w. Comparison between the groups post-intervention: The HILT + EX group had a larger decrease in pain than the SL + EX group, with the smallest effect experienced by the HILT group, at both 4 w and 12 w. HILT + EX showed a higher improvement in functional outcomes than SL + EX, while no significant difference was found between SL + EX and HILT alone.
Basford, 1999 [38]	61 31/28	48 HILT, 48 SL	(1) HILT (2) SL **Comparator:** SL	VAS, ODI	Pre, Post, 4 wFU	Significant improvements in pain and function were observed in both groups. Comparison between the groups post-intervention: The HILT group scored significantly better in comparison to the SL group for pain and function.
Choi, 2017 [39]	20 Nr	48 HILT, 47 CPT	(1) HILT+ CPT (2) CPT **Comparator:** CPT: 20 min; hot pack, 15 min. interference wave and 5 min; deep heat injection using ultrasonic waves 3 times per week for 4 weeks.	VAS, ODI	Pre, Post	Significant improvements in pain and function were observed in both groups. Comparison between the groups post-intervention: The HILT + CPT group scored significantly better in comparison to the CPT group for both pain and function.
Conte, 2009 [40]	56 Nr	Nr (range 18–65)	(1) HILT + EX (2) EX **Comparator**: EX: back school (upper and lower limb stretches, Klapp kneeling position, costal and diaphragm ventilation, muscle strengthening, exercises in front of the mirror to find neutral posture)	VAS, MODQ	Pre, Post	Significant improvements in pain and function were observed in both groups. Comparison between the groups post-intervention: The HILT + EX group scored significantly better in comparison to the EX group for pain and function.
Fiore, 2011 [42]	30 11/19	51	(1) HILT (2) US **Comparator**: US: 2 W/cm^2^ for 10 min.	VAS, ODI	Pre, Post	Significant improvements in pain and function were observed in both groups. Comparison between the groups post-intervention: The HILT group scored significantly better in comparison to the US group for pain and function.
Gocevska, 2019 [43]	54 29/25	55 HILT	(1) HILT + EX (2) US + EX **Comparator**: US: 0.5 W/cm^2^ for 5 min.	NRS, ODI	Pre, Post, 12 wFU	Significant improvements in pain and function were observed in both groups post-treatment and the results remained consistent at 12 w. Comparison between the groups post-intervention: The HILT + EX group scored significantly better in comparison to US + EX for pain and function at both post-treatment and 12 w.
**Neck Pain**						
Alayat et al., 2016 [36]	60 60/0	36 HILT, 25 SL	(1) HILT + EX (2) SL + EX Comparator: SL	VAS, NDI	Pre, Post	Significant improvements in pain and function were observed in both groups. Comparison between the groups post-intervention: The HILT + EX group scored significantly better in comparison to the SL + EX group for both pain and function.
Alayat, 2017 [37]	75 75/0	46	(1) HILT (MLS) +EX (2) LLLT + EX (3) SL + EX **Comparators**: (1) SL (2) LLLT Infrared, 830 nm laser with 800 mW power, 30 min/session, delivering 300 J	VAS, NDI	Pre, Post	Significant improvements in pain and function were observed in all 3 groups. Comparison between the three groups post-intervention: The greatest improvement in both pain and function was seen in the HILT (MLS) + EX group, followed by LLLT + EX, then SL + EX.
**Myofascial Pain Syndrome**						
Dundar, 2015 [41]	76 0/76	40 HILT 38 SL	(1) HILT + EX (2) SL + EX **Comparator**: SL	VAS, NDI	Pre, 1 wFU, 9 wFU	Significant improvements in pain and function were observed in both groups. Comparison between the groups post-intervention: The HILT + EX group showed a greater improvement in pain and function than the SL + EX group.
**Myogenic Temporomandibular Joint Disorder**						
Ekici et al., 2021 [47]	76 Nr	32 HILT 30 SL	(1) HILT (2) SL **Comparator**: SL	VAS, JFLS-20	Pre, 1 wFU, 9 wFU	Significant improvements in pain and function were observed in both groups. Comparison between the groups post-intervention: Percentage changes yielded a significantly greater improvement in pain and function in the HILT group in comparison to the SL group.

Legend: CPT: Conservative physiotherapy treatment; EMF: Pulsed electromagnetic field; EX: Therapeutic Exercises; HILT: High-intensity laser therapy; LLLT: Low-level laser therapy; JFLS-20: Jaw Functional Limitation Scale 20; MLS: Multiwave Locked System; MODQ: Modified Oswestry Disability Questionnaire; MT: Medical therapy agents; NDI: Neck disability index; ODI; Oswestry Disability Index; PDI: Pain disability Index; Post: After the treatment; Pre: Before the treatment; RDQ: Roland disability questionnaire; SL: Sham laser (placebo laser); US: Ultrasound treatment; VAS: visual analogue scale; NRS: Numeric rating scale; wFU: weeks of follow-up; yFU: years of follow-up; Nr: not reported.

**Table 3 jcm-11-03701-t003:** Characteristics of the laser parameters in included studies.

Study	Type of Laser Wavelength (nm) Application Mode	Peak Power P_peak_ (Pulsed); N/A for CW	Frequency (Hz)/ Pulse Duration (µs)	Energy per Pulse (E) P_peak_ × t (laser Pulse Duration)	Energy Dose (J) per Point/ per All Points/ per Scanning/ per Treatment/ Accumulated Energy from All Sessions (J)	Treatment Time per Point/ per ST (All Points)/ per SC per Session/ no. Sessions/ no. Sessions per Week/Total No. Weeks	Average Power (Pulsed) or Output Power (CW)/ Scanning Phase	Average Power (Pulsed) or Output Power (CW) ST per All Points/ ST per Point	Area (cm^2^)/ Spot Size (cm^2^)	Average Power Density (mW/cm^2^) SC Scanning	Average Power Density (mW/cm^2^) ST per All Points/ ST per Point	Dose Density (J/cm^2^) SC	Dose Density (J/cm^2^) ST per All Points/ ST per Point
Leclair et al., 2007 [33]	Pulsed KTP-Nd:YAG 532 Nr	Nr	Nr/ 15 ms	Nr	Nr	2.84 sessions (range 1–8)	Nr	Nr	Nr 0.4	Nr	Nr	10 (Nr if SC or ST)	
Abdelbasset et al., 2020 [44]	Pulsed- Nd:YAG 1064 SC	Nr	Nr/Nr	Nr	N/A/ N/A/ 300 first 2 weeks 120–150 next 4 weeks/ **3240–3600**	N/A/ N/A/ 75 s first 2 weeks 30 s next 4 weeks/ 18 sessions 3 x/week 6 weeks	(6–12 W) 8 W first 2 weeks 6 W next 4 weeks	N/A	30 N/A	**200–266.7**	N/A	6	N/A
Abdelbasset et al., 2020 [43]	**Pulsed** Ga-Ar 1064 SC	Nr	Nr/Nr	Nr	N/A/ N/A/ 1200/ **2800**	N/A/ N/A/ 15 min 24 sessions 2 x/week 12 weeks	(12 W) **1.33**	N/A	Nr/ N/A	Nr	N/A	150	N/A
Abdelbasset et al., 2021 [45]	Nd: YAG 1064 nm Pulsed B	3 kW	10–40/ 120	0.35 J	25/ 200/ 2800/ **3000/** **48,000**	14 s/ **112 s/** **788 s/** 15 min. 16 sessions 2 x/week 8 weeks	(10.5 W) **3.55** SC	**1.79 W ST all points/** **1.79 W ST per point**	60 0.2	220	**1118 ST** **8950 ST point**	0.61; 0.71; 0.81 SC	**125 ST all points/** 0.61 ST per point
Alayat et al., 2014 [34]	Pulsed- Nd:YAG 1064 B	3 kW	10–40/ 120–150	**0.36 J–** **0.45 J**	25/ 200/ 2800/ 3000/ **36,000**	14 s/ **112 s**/ **788 s**/ 15 min. 12 sessions 3 x/week 4 weeks	**(3.6–18 W)** **3.55 SC**	**1.79 W ST all points/** **1.79 W ST per point**	Nr 0.2	? SC	**1118 ST** **8950 ST point**	0.61; 0.71; 0.81 SC	**125 ST all points/** 0.61 ST per point
Alayat et al., 2016 [35]	Pulsed- Nd:YAG 1064 B	3 kW	10–40/ 120–150	**0.36 J–** **0.45 J**	25/ 200/ 2050/ 2250/ **27,000**	14 s/ **112 s**/ **788 s**/ 15 min. 12 sessions 2 x/weeks 6 weeks	**(3.6–18 W)** **2.6 SC**	**1.79 ST all points/** **1.79 ST per point**	75 0.2	**34.7 SC**	**1118 ST** **8950 ST point**	27.3 SC	**125 ST all points/** 0.51 ST per point
Alayat et al., 2017 [36]	Pulsed and CW Ga-Al-Ar 808–905 B	1000 mW (CW) 25 W (Pulsed) Continuous and pulsed synchronously	N/A for CW 1500/Nr	**Nr**	**12.6/** **100.8/** 300/ **400.8/** **801.6**	30 s/ **240 s/** 256 s/ **8 min 16 s/** 12 sessions/ 2 x/week 6 weeks	0.5 W CW 0.054 W (54 mW)Pulsed Nr SC	0.054–0.5 all points/ 0.054–0.5 per point	75 3.14	**0.72–6.67 SC**	**2.15–19.9 ST** **17.2–159.24 ST point**	4 SC	4 ST (Nr if per all points or per point)
Basford et al., 1999 [37]	CW- Nd:YAG 1060 ST	N/A	N/A	N/A	**239/** **1912/** N/A **1912/** **22,944**	90 s/ 360 s/ N/A 6 min. 12 sessions 3 x/weeks 4 weeks	(N/A–cont.) N/A	**5.31 W ST all points/** **2.66 W ST per point**	N/A 4.9	N/A	**14.75 ST** 542 ST point	N/A	**48.77 ST all points/** **48.9 ST per point**
Choi et al., 2017 [38]	Nr 1064 Nr Nr	Nr	Nr/Nr	Nr	Nr/ Nr	10 min. 12 sessions 3 x/weeks 4 weeks	Nr	Nr	Nr	Nr	Nr	1.378 (Nr if SC or ST)	
Conte et al., 2009 [39]	Pulsed Nd:YAG Nr B	Nr	Nr/Nr	Nr/Nr	18–20/ **72–80**/ 3000/ **3071–3080**/ Nr/	Nr	Nr	Nr/Nr	Nr Nr	Nr	Nr Nr	0.66; 0.71; 0.76 SC	0.66 (Nr if per point or all points)
Dundar et al., 2015 [40]	Pulsed- Nd:YAG 1064 B	3.8 kW	10–40/ 120–150	**0.456 J–** **0.57 J**	10/ 60/ 1000/ 1060/ **15,900**	6 s/ **18 s**/ **882 s**/ 15 min. 15 sessions 5 x/weeks 3 weeks	**(4.56–22.8 W)** **1.13 SC**	**3.33 ST all points** **1.67 ST per point**	100 0.2	**11.3 SC**	**2775 ST** **8 350 ST point**	0.36; 0.41; 0.51 SC	**50 ST all points/** 0.61 ST per point
Ekici et al., 2021 [46]	Pulsed Nd:YAG 1064 B	3 kW	10–40/ 120–150	**0.36 J–** **0.45 J**	5.51/ 33.1/ 996/ 1029.2/ **15,438**	6 s/ **36 s/** **864 s/** 15 min 15 sessions 5 x/week 3 weeks	10.5 W **1.15 SC**	**0.92 ST all points** **0.92 ST per point**	100 0.2	**11.5 SC**	**575 ST** **4600 ST point**	0.36; 0.41; 0.51 SC	**27.6 ST all points/** 0.61 ST per point
Fiore et al., 2011 [41]	Pulsed- Nd:YAG 1064 B	1 kW	Nr/ less than 150	Nr	Nr/ Nr/ Nr/ 2600/ **39,000**	Nr/ Nr/ Nr/ 10 min. 15 sessions 5 x/weeks 3 weeks	(6 W) ? SC	? ST all points/ ? ST per point	100 0.2	Nr	Nr	0.71 (for all treatments Nr values of SC and ST phase)	
Gocevska et al., 2019 [42]	CW Nr **940 nm** SC	Nr	Nr/ Nr	N/A	N/A/ N/A/ 2400/ 2400/ **24,000**	N/A/ N/A/ 15 min/ 15 min 10 sessions 5 x/weeks 2 weeks	4 W	N/A	Nr N/A	Nr	N/A	1.5 SC	N/A

Legend: SC: Scanning mode; ST: Stationary (point-to-point) mode; B: Both SC and ST modes; CPT: Conservative physiotherapy treatment; EX: Therapeutic Exercises; HILT: High-intensity laser therapy; MLS: Multiwave Locked System; MODQ: Modified Oswestry Disability Questionnaire; MPS: Myofascial pain syndrome; MSK: Musculoskeletal; MT: Medical therapy agents; NDI: Neck disability index; NS-CLBP: Non-specific low back pain; NP: Neck pain; ODI; Oswestry Disability Index; Post: After the treatment; Pre: Before the treatment; RDQ: Roland disability questionnaire; SL: Sham laser (placebo laser); US: Ultrasound treatment; VAS: visual analogue scale; wFu: weeks of follow-up. Entries in **bold** were not reported/unavailable and were calculated or assumed by our team.

**Table 4 jcm-11-03701-t004:** Characteristics of the Cohen’s d effect sizes and 95% CIs for all time points for the included studies. Cohen’s d effect sizes were interpreted as weak if between 0.2 and 0.49, medium if between 0.5 and 0.79, large if between 0.8 and 1.19, very large if between 1.2 and 1.99, and huge if equal or higher than 2.

Studies	Comparison Groups Group 1 vs. Group 2	Outcome Measures	Time Point	Experimental Group 1 Mean ± SD	Comparator Group 2 Mean ± SD	Effect Size	95% Confidence Interval
Abdelbasset et al., 2020 [44]	HILT + EX vs. SL + EX	VAS	Pre	6.7 ± 1.6	6.9 ± 1.5	N/A	N/A
	HILT + EX vs. SL + EX	VAS	Post	3.7 ± 1.1	6.1 ± 1.3	**1.998**	**1.147–2.758**
	HILT + EX vs. SL + EX	ODI	Pre	39.8 ± 14.3	38.6 ± 12.9	N/A	N/A
	HILT + EX vs. SL + EX	ODI	Post	19.3 ± 6.7	35.4 ± 11.5	**1724**	**0.913–2.456**
	HILT + EX vs. SL + EX	PDI	Pre	33.5 ± 10.7	34.3 ± 11.2	N/A	N/A
	HILT + EX vs. SL + EX	PDI	Post	24.7 ± 7.6	30.8 ± 9.8	0.698	0.000–1.364
Abdelbasset et al., 2020 [43]	HILT + EX vs. LLLT + EX	VAS	Pre	6.3 ± 1.9	6.5 ± 1.7	N/A	N/A
	HILT + EX vs. LLLT + EX	VAS	Post	3.5 ± 0.8	3.4 ± 0.9	0.117	−0.505 to 0.735
	HILT + EX vs. LLLT + EX	ODI	Pre	37.3 ± 11.3	36.5 ± 12.7	N/A	N/A
	HILT + EX vs. LLLT + EX	ODI	Post	18.5 ± 7.2	17.8 ± 6.4	0.103	−0.519 to 0.721
	HILT + EX vs. EX	VAS	Pre	6.3 ± 1.9	6.6 ± 1.6	N/A	N/A
	HILT + EX vs. EX	VAS	Post	3.5 ± 0.8	5.9 ± 1.8	**1.723**	**0.967–2.411**
	HILT + EX vs. EX	ODI	Pre	37.3 ± 11.3	36.2 ± 12.3	N/A	N/A
	HILT + EX vs. EX	ODI	Post	18.5 ± 7.2	34.6 ± 11.8	**1.852**	**1.078–2.551**
Abdelbasset et al., 2021 [45]	HILT + EX vs. EMF + EX	VAS	Pre	7.4 ± 2.2	7.2 ± 1.9	N/A	N/A
	HILT + EX vs. EMF + EX	VAS	Post	3.2 ± 1.2	5.1 ± 1.7	1.291	0.525–1.997
	HILT + EX vs. EMF + EX	MODQ	Pre	42.4 ± 12.7	41.8 ± 11.8	N/A	N/A
	HILT + EX vs. EMF + EX	MODQ	Post	23.6 ± 6.5	29.3 ± 8.7	0.742	0.031–1.419
	HILT + EX vs. EMF + EX	PDI	Pre	34.8 ± 11.4	34.5 ± 10.9	N/A	N/A
	HILT + EX vs. EMF + EX	PDI	Post	22.6 ± 7.2	27.2 ± 9.5	0.546	−0.151 to 1217
	HILT + EX vs. EX	VAS	Pre	7.4 ± 2.2	6.8 ± 2.1	N/A	N/A
	HILT + EX vs. EX	VAS	Post	3.2 ± 1.2	6.3 ± 1.8	2.027	1.158–2.799
	HILT + EX vs. EX	MODQ	Pre	42.4 ± 12.7	40.5 ± 12.3	N/A	N/A
	HILT + EX vs. EX	MODQ	Post	23.6 ± 6.5	35.7 ± 10.6	1.376	0.599–2.088
	HILT + EX vs. EX	PDI	Pre	34.8 ± 11.4	33.9 ± 10.7	N/A	N/A
	HILT + EX vs. EX	PDI	Post	22.6 ± 7.2	30.5 ± 10.2	0.895	0.170–1.577
Alayat et al., 2014 [34]	HILT + EX vs. SL + EX	VAS	Pre	8.36 ± 0.95	8.21 ± 1.1	N/A	N/A
	HILT + EX vs. SL + EX	VAS	Post	2.04 ± 0.83	3.21 ± 0.83	**1.410**	**0.801–2.018**
	HILT + EX vs. SL + EX	VAS	12 wFu	2.64 ± 1.25	3.71 ± 1.30	**0.840**	**0.272–1.409**
	HILT + EX vs. HILT	VAS	Pre	8.36 ± 0.95	8.35 ± 0.88	N/A	N/A
	HILT + EX vs. HILT	VAS	Post	2.04 ± 0.83	4.15 ± 2.03	**1.454**	**0.81–2.097**
	HILT + EX vs. HILT	VAS	12 wFu	2.64 ± 1.25	5.65 ± 1.04	**2.577**	**1.806–3.349**
	HILT + EX vs. SL + EX	RDQ	Pre	15.46 ± 1.17	15.63 ± 1.56	N/A	N/A
	HILT + EX vs. SL + EX	RDQ	Post	4.43 ± 1.28	5.75 ± 0.99	**1.142**	**0.554–1.73**
	HILT + EX vs. SL + EX	RDQ	12 wFu	5.5 ± 1.17	6.92 ± 0.78	**1.407**	**0.798–2.015**
	HILT + EX vs. HILT	RDQ	Pre	15.46 ± 1.17	15.4 ± 1.19	N/A	N/A
	HILT + EX vs. HILT	RDQ	Post	4.43 ± 1.28	6.35 ± 1.6	**1.351**	**0.717–1.986**
	HILT + EX vs. HILT	RDQ	12 wFu	5.5 ± 1.17	7.35 ± 1.5	**1.405**	**0.766–2.044**
	HILT + EX vs. SL + EX	MODQ	Pre	34.11 ± 3.14	34.5 ± 2.93	N/A	N/A
	HILT + EX vs. SL + EX	MODQ	Post	13.9 ± 3.83	16.41 ± 3.01	0.722	0.159–1.285
	HILT+EX vs. SL+EX	MODQ	12 wFu	15.14 ± 4.3	18.75 ± 3.07	**0.954**	**0.379–1.529**
	HILT+EX vs. HILT	MODQ	Pre	34.11 ± 3.14	35.55 ± 3.62	N/A	N/A
	HILT+EX vs. HILT	MODQ	Post	13.9 ± 3.83	17.25 ± 3.14	**0.941**	**0.337–1.545**
	HILT+EX vs. HILT	MODQ	12 wFu	15.14 ± 4.3	19.05 ± 2.96	**1.028**	**0.418–1.637**
Alayat et al., 2016 [35]	HILT + EX vs. SL + EX	VAS	Pre	8.00 ± 0.79	7.83 ± 0.80	N/A	N/A
	HILT + EX vs. SL + EX	VAS	Post	1.77 ± 0.73	2.83 ± 0.79	**1.394**	**0.829–1.958**
	HILT + EX vs. SL + EX	NDI	Pre	45.87 ± 5.12	47.97 ± 3.29	N/A	N/A
	HILT + EX vs. SL + EX	NDI	Post	7.80 ± 1.65	9.86 ± 1.48	**1.314**	**0.756–1.872**
Alayat et al., 2017 [36]	HILT(MLS) + EX vs. LLLT + EX	VAS	Pre	39.76 ± Nr	37.88 ± Nr	N/A	N/A
	HILT(MLS) + EX vs. LLLT + EX	VAS	Post	19.58 ± Nr	38.90 ± Nr	Insufficient data	Insufficient data
	HILT(MLS) + EX vs. LLLT + EX	NDI	Pre	37.80 ± Nr	36.08 ± Nr	N/A	N/A
	HILT (MLS) + EX vs. LLLT + EX	NDI	Post	17.82 ± Nr	37.18 ± Nr	Insufficient data	Insufficient data
	HILT(MLS) + EX vs. SL + EX	VAS	Pre	39.76 ± Nr	36.36 ± Nr	N/A	N/A
	HILT(MLS) + EX vs. SL + EX	VAS	Post	19.58 ± Nr	55.52 ± Nr	**2.223 ***	−1.303 to 5.748 *
	HILT(MLS) + EX vs. SL + EX	NDI	Pre	37.80 ± Nr	40.12 ± Nr	N/A	N/A
	HILT(MLS) + EX vs. SL + EX	NDI	Post	17.82 ± Nr	59.00 ± Nr	**2.63 ***	−1.155 to 6.416 *
Basford et al., 1999 [37]	HILT vs. SL	VAS	Pre	35.2 ± Nr	37.4 ± Nr	N/A	N/A
	HILT vs. SL	VAS	Post	17.1 ± Nr	32.8 ± Nr	Insufficient data	Insufficient data
	HILT vs. SL	VAS	4 wFu	19.1 ± Nr	35.1 ± Nr	Insufficient data	Insufficient data
	HILT vs. SL	ODI	Pre	21 ± Nr	26 ± Nr	N/A	N/A
	HILT vs. SL	ODI	Post	13.3 ± Nr	22.6 ± Nr	Insufficient data	Insufficient data
	HILT vs. SL	ODI	4 wFu	14.7 ± Nr	22.9 ± Nr	Insufficient data	Insufficient data
Choi et al., 2017 [38]	HILT + CPT vs. CPT	VAS	Pre	7.0 ± 0.8	7.0 ± 0.8	N/A	N/A
	HILT + CPT vs. CPT	VAS	Post	3.4 ± 0.8	6.2 ± 1.4	**2.714**	**1.69–3.738**
	HILT+CPT vs. CPT	ODI	Pre	31.6 ± 11.5	33.1 ± 13.0	N/A	N/A
	HILT+CPT vs. CPT	ODI	Post	19.0 ± 10.6	29.6 ± 10.7	**0.997**	**0.197–1.797**
Conte et al., 2009 [39]	HILT + EX vs. EX	VAS	Pre	60 ± 19.5	63.32 ± 16.8	N/A	N/A
	HILT + EX vs. EX	VAS	Post	27.9 ± 15	45.3 ± 14.3	**1.187**	**0.619–1.755**
	HILT + EX vs. EX	MODQ	Pre	21.39 ± 6.9	23.12 ± 6.98	N/A	N/A
	HILT + EX vs. EX	MODQ	Post	9.6 ± 5.98	16.6 ± 7.38	**1.042**	**0.484–1.6**
Dundar et al., 2015 [40]	HILT + EX vs. SL + EX	VAS at rest	Pre	5.9 ± 1.4	5.7 ± 1.5	N/A	N/A
	HILT + EX vs. SL + EX	VAS at rest	1 wFu	2.7 ± 1.2	4.2 ± 1.6	**1.063**	**0.579–1.546**
	HILT + EX vs. SL + EX	VAS at rest	9 wFu	2.6 ± 1.2	4.1 ± 1.4	**1.152**	**0.663–1.64**
	HILT + EX vs. SL + EX	NDI	Pre	32.6 ± 6.6	32.9 ± 8.3	N/A	N/A
	HILT + EX vs. SL + EX	NDI	1 wFu	21.1 ± 6.3	26.6 ± 7.1	**0.82**	**0.349–1.291**
	HILT + EX vs. SL + EX	NDI	9 wFu	20.3 ± 6.22	26.1 ± 6.7	**0.898**	**0.423–1.373**
Ekici et al., 2021 [46]	HILT vs. SL	VAS	Pre	60.9 ± 21.9	59.3 ± 20.5	N/A	N/A
	HILT vs. SL	VAS	1 wFu	27.7 ± 19	56.8 ± 19.6	**1.51**	**0.96–2.02**
	HILT vs. SL	VAS	9 wFu	26.3 ± 24	55 ± 18.8	**1.33**	**0.8–1.83**
	HILT vs. SL	JFLS-20	Pre	72.15 ± 47.16	53.50 ± 33.86	N/A	N/A
	HILT vs. SL	JFLS-20	1 wFu	Nr	Nr	Insufficient data	Insufficient data
	HILT vs. SL	JFLS-20	9 wFu	Nr	Nr	Insufficient data	Insufficient data
Fiore et al., 2011 [41]	HILT vs. US	VAS	Pre	7 ± Nr	7 ± Nr	N/A	N/A
	HILT vs. US	VAS	Post	3 ± Nr	4 ± Nr	Insufficient data	Insufficient data
	HILT vs. US	ODI	Pre	28 ± Nr	28 ± Nr	N/A	N/A
	HILT vs. US	ODI	Post	12 ± Nr	16 ± Nr	Insufficient data	Insufficient data
Gocevska et al., 2019 [42]	HILT + EX vs. US + EX	NRS	Pre	7.22 ± 8.85	6.96 ± 0.94	N/A	N/A
	HILT + EX vs. US + EX	NRS	Post	2.11 ± 0.8	4.26 ± 1.06	**2.290**	**1.603–2.976**
	HILT + EX vs. US + EX	NRS	12 wFU	1.89 ± 0.64	4.89 ± 0.85	**3.987**	**3.065–4.909**
	HILT + EX vs. US + EX	ODI	Pre	44.33 ± 3.92	45.22 ± 3.91	N/A	N/A
	HILT + EX vs. US + EX	ODI	Post	16.29 ± 4.85	26.74 ± 4.51	**3.987**	**1.271–3.192**
	HILT + EX vs. US + EX	ODI	12 wFU	15.89 ± 4.58	26.63 ± 3.73	**2.571**	**1.85–3.292**

Legend: CPT: Conservative physiotherapy treatment; EX: Therapeutic Exercises; HILT: High-intensity laser therapy; JFLS-20: Jaw Functional Limitation Scale-20; MLS: Multiwave Locked System; MODQ: Modified Oswestry Disability Questionnaire; MT: Medical therapy agents; NDI: Neck disability index; ODI; Oswestry Disability Index; Post: After the treatment; PDI: Pain Disability Index; Pre: Before the treatment; RDQ: Roland disability questionnaire; SD: standard deviation; SL: Sham laser (placebo laser); US: Ultrasound treatment; VAS: visual analogue scale; wFu: weeks of follow-up. Nr: not reported; LLLT: Low-level laser therapy. Entries in **bold** are large, very large, and huge effect sizes. * values retrieved from the source article.

## Data Availability

The data used to support the findings of this study are available from the corresponding author by email (Melanie Morin, melanie.m.morin@usherbrooke.ca) upon request.

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
