# Peer review of "High-Intensity Laser Therapy (HILT) as an Emerging Treatment for Vulvodynia and Chronic Musculoskeletal Pain Disorders: A Systematic Review of Treatment Efficacy"

_jcm, 2022, doi:10.3390/jcm11133701_

Round 1
Reviewer 1 Report
The submitted paper was well written with an acceptable methodology. There are some questions and suggestions about the current format:
1. You bolded the "vulvodynia" in the title, however, only one related article was found so you can not claim that you systemically review this topic.
2. In the results, in dosimetry section please write the range of parameters used in the reviewed articles.
Author Response
Please find below our answers to the reviewer’s 1 comments in addition to the modified manuscript. We appreciate and thank the reviewers for their comments and feel that the manuscript in its present state is stronger with their input. We hope you will appreciate this new version.
The submitted paper was well written with an acceptable methodology. There are some questions and suggestions about the current format:
- You bolded the "vulvodynia" in the title, however, only one related article was found so you can not claim that you systemically review this topic.
We appreciate the reviewer’s opinion, and we acknowledge that we found only one study on vulvodynia. However, we have demonstrated the current state of the evidence through a rigorous and systematic search. Even though we only included one low-quality trial on vulvodynia, showing that is a valid effect - our systematic review has made a significant contribution to the field as it clearly demonstrates the lack of evidence for this condition. We believe it is crucial, especially considering the warning statement issued by the FDA and leading scientific societies (mentioned in lines 82-85). Therefore, we decided to keep the “vulvodynia” in the review title as we believe that demonstrating the low level of existing evidence in that area, based on a sound systematic search, is an important contribution to guiding research and clinical practice.
- In the results, in dosimetry section please write the range of parameters used in the reviewed articles.
We thank the reviewer for this suggestion. The manuscript has been revised accordingly.

Reviewer 2 Report
It is an interesting study on a problem of great interest, because the therapy is not very effective and the discomfort is very demanding. Studies like this one are necessary to shed light on therapies that are not studied with the necessary objectivity. The fact that it is not correct does not eliminate its usefulness, but it teaches us that we should be cautious when recommending it.
The introduction sets out very interesting aspects to introduce and raise awareness of the subject of the review. The method and the inclusion criteria seem to me to be very appropriate. The decision to discriminate the flowcharts according to the objective seems to me to be appropriate.
Minor comment
The aim of the study is really deep, but a section on the objective of the review is missing where the intention of the review is clear. It moves from the introduction to the methodology and the objective is intuited. Explained at the end of last paragraph. “The aim of this systematic review was thus to systematically locate, critically appraise, and synthesize the available evidence on the effectiveness of HILT in reducing pain and improving function in (1) women with vulvodynia and (2) patients with chronic primary musculo-skeletal pain. Moreover, laser parameters were also retrieved, and effect sizes calculated, whenever possible, to discuss their relative effects.”
I presume that table 3 a,3 b, and figure 3 can be more visual under a figure of Risk-of-bias summary (The Cochrane risk-of-bias tool (Review manager version 5.4))
Author Response
Please find enclosed a Word document with our answers to the reviewer’s 2 comments in addition to the modified manuscript. We appreciate and thank the reviewers for their comments and feel that the manuscript in its present state is stronger with their input. We hope you will appreciate this new version.
It is an interesting study on a problem of great interest, because the therapy is not very effective and the discomfort is very demanding. Studies like this one are necessary to shed light on therapies that are not studied with the necessary objectivity. The fact that it is not correct does not eliminate its usefulness, but it teaches us that we should be cautious when recommending it.
The introduction sets out very interesting aspects to introduce and raise awareness of the subject of the review. The method and the inclusion criteria seem to me to be very appropriate. The decision to discriminate the flowcharts according to the objective seems to me to be appropriate.
Minor comment
The aim of the study is really deep, but a section on the objective of the review is missing where the intention of the review is clear. It moves from the introduction to the methodology and the objective is intuited. Explained at the end of last paragraph. “The aim of this systematic review was thus to systematically locate, critically appraise, and synthesize the available evidence on the effectiveness of HILT in reducing pain and improving function in (1) women with vulvodynia and (2) patients with chronic primary musculo-skeletal pain. Moreover, laser parameters were also retrieved, and effect sizes calculated, whenever possible, to discuss their relative effects.”
We thank the reviewer for the words of appreciation for our work. According to the reviewer's comment, we have created a separate, “objectives” section with the aims of our systematic review. However, that is not compatible with the template provided by the JCM Editorial Office therefore we may need the acceptance of the proof-editor.
I presume that table 3 a,3 b, and figure 3 can be more visual under a figure of Risk-of-bias summary (The Cochrane risk-of-bias tool (Review manager version 5.4))
We agree with the reviewer’s comment. The tables have been modified as suggested.
